# In Situ Assembly of Nanomaterials and Molecules for the Signal Enhancement of Electrochemical Biosensors

**DOI:** 10.3390/nano11123307

**Published:** 2021-12-06

**Authors:** Yong Chang, Ning Xia, Yaliang Huang, Zhifang Sun, Lin Liu

**Affiliations:** College of Chemistry and Chemical Engineering, Anyang Normal University, Anyang 455000, China; 7180610011@stu.jiangnan.edu.cn (Y.C.); xianing82414@csu.edu.cn (N.X.); 182301010@csu.edu.cn (Y.H.)

**Keywords:** electrochemical biosensors, self-assembly, nanomaterials, hybridization, peptide, streptavidin

## Abstract

The physiochemical properties of nanomaterials have a close relationship with their status in solution. As a result of its better simplicity than that of pre-assembled aggregates, the in situ assembly of nanomaterials has been integrated into the design of electrochemical biosensors for the signal output and amplification. In this review, we highlight the significant progress in the in situ assembly of nanomaterials as the nanolabels for enhancing the performances of electrochemical biosensors. The works are discussed based on the difference in the interactions for the assembly of nanomaterials, including DNA hybridization, metal ion–ligand coordination, metal–thiol and boronate ester interactions, aptamer–target binding, electrostatic attraction, and streptavidin (SA)–biotin conjugate. We further expand the range of the assembly units from nanomaterials to small organic molecules and biomolecules, which endow the signal-amplified strategies with more potential applications.

## 1. Introduction

During the past several decades, miscellaneous nanomaterials with various elements and different morphologies have been designed and synthesized. Owing to the amazing and powerful properties, they have been widely utilized in chemical, physical, and biological-related fields. Generally, the properties of nanomaterials are closely related to their physicochemical parameters, including composition, shape, and size. For instance, silver nanoparticles (AgNPs) can generate a well-defined and amplified electrochemical peak based on the highly characteristic solid-state Ag/AgCl process [1], which have been widely used as the electrochemical tracers for the detection of various targets [2]. The status of nanomaterials in solution (monodispersion and aggregation) may have an important influence on their performances. For example, gold nanoparticles (AuNPs) and AgNPs exhibit different local surface plasmon resonance (LSPR) adsorption and endow the solution with different color. The inherent enzyme-mimetic catalytic activity can also be reversibly regulated by modulating the status of nanozymes, such as MoS_2_, quantum dots (QDs), Cu_2−x_Se NPs, and AuNPs [3,4,5,6]. Traditional fluorescent dyes show aggregation-induced quenching properties, but on the contrary, aggregation-induced emission (AIE) phenomenon was also observed in various organic molecules and nanomaterials [7,8,9,10]. Therefore, lots of optical methods (fluorescence and colorimetric) have been precisely explored to analyze targets of interest based on the aggregation-induced effect, including metal ions, small molecules, DNA, and enzymes [11,12]. Up to now, several mechanisms for stimulating the aggregation of nanomaterials have been demonstrated, including DNA hybridization, antigen–antibody association, aptamer–target binding, electrostatic attraction, streptavidin (SA)–biotin interaction, metal ion–ligand coordination, and covalent bond formation [13,14,15,16,17,18]. However, the homogeneous methods based on target-recognition-induced aggregation in solution are less sensitive.

As a result of the merits of low cost, high sensitivity, and ease of calibration and miniaturization, electrochemical biosensors have received broad research interest for applications from environmental monitoring to food safety and disease diagnosis. To further improve their sensitivity, plenty of signal-amplified strategies have been designed and applied in electrochemical assays. Usually, the signal-amplified strategies contain enzyme catalysis (e.g., horseradish peroxidase and alkaline phosphatase), DNA assembly techniques such as catalytic hairpin assembly (CHA) and hybridization chain reaction (HCR), and various functional nanomaterials [19,20,21]. Among these protocols, nanomaterials have shown great promise for improving the sensitivity and selectivity of electrochemical assays because of their excellent characteristics, such as good conductivity, high surface to volume ratio, and ease of functionalization. Up to now, nanomaterials, including carbon nanotubes, graphene, semiconductor quantum dots (QDs), metal nanoparticles (NPs), and 2D layered nanosheets, have been exploited for the fabrication of electrochemical biosensors with enhanced performances [22]. During various electrochemical assays, nanomaterials mainly play four important roles for enhancing the detection performances: electrode substrate modifiers, nanoelectrocatalysts, nanocarriers for enzymes and recognition elements, and electroactive tracers. For example, carbon nanomaterials are frequently utilized in electroanalytical and electrocatalytic sensing fields because of their high electrical conductivity, excellent surface to volume ratio, and chemical stability [23]. Gold nanoparticles (AuNPs) are easily incorporated with enzymes and biorecognition elements, perfectly combining the catalytic properties of enzymes, recognition abilities of biorecognition elements with excellent electrochemical properties of AuNPs [24].

It is a promising signal-amplified strategy to integrate the organized assembly or unordered aggregation of nanomaterials into electrochemical methods [25]. Moreover, the excellent conductivity and electrocatalytic activity of nanomaterials can significantly enhance the conductivity and redox current. For instance, Chen et al. employed melamine to induce the formation of the PdPt nanodendrites–melamine networks based on the firm interactions between the nanodendrites and the three amino groups of each melamine molecule [26]. Then, the formed networks with excellent catalytic ability were utilized as the labels to increase the current for gene mutation detection. AgNPs aggregates induced by the hybridization of DNA on AgNPs were successfully applied for multiplexed DNA target detection [27]. However, these aggregate tags were synthesized through hybridization prior to the sandwich assays, which may affect the uniformity of the size of the aggregates. In 2015, Dai et al. introduced the concept of re-creation of the existing platforms, which transferred the NPs-based colorimetric assay into the electrochemical analysis with Hg^2+^ as the model analyte [28]. After that, a number of attempts have been put into the construction of various detection techniques based on the conversion of aggregation-based colorimetirc assays to interfacial analytical assays [29,30,31,32,33]. In addition to acting as the aggregation triggers, biomacromolecules with nanoscale sizes can also be used as the self-assembling building blocks to form diverse nanostructures for drug delivery and biosensing. Meanwhile, the low conductivity of biomacromolecules facilitates the development of electrochemical impedance biosensors.

In this review, we mainly discuss the design and application of electrochemical biosensors based on the in situ assembly of nanomaterials on the electrode surface for signal readout and amplification. To facilitate the readability and comprehension, the works are briefly discussed based on the difference in the interactions for the assembly of nanomaterials. Furthermore, we expand the range of assembly units from nanomaterials to small molecules and biomacromolecules. 

## 2. In Situ Assembly of Nanomaterials for Signal Amplification

### 2.1. DNA Hybridization

Due to the specificity of base-pairing hybridization, a variety of DNA assembly nanotechnologies have been elaborately designed and versatilely applied in biosensing for signal amplification [34,35]. It is a promising signal amplification strategy to combine the assembly technologies with nanomaterials. Moreover, the excellent conductivity and electrocatalytic activity of nanomaterials can significantly transform the intrinsic electron inert polymeric DNA into a conductive DNA nanostructure. Among them, hybridization-inducing aggregation without the use of enzymes is simple and can be conducted at mild conditions [36]. Song et al. developed a disposable electrochemical aptasensor array for multiplied proteins detection by in situ DNA hybridization-induced AgNPs aggregation for signal amplification [37]. As shown in Figure 1A, AgNPs were modified with two complementary DNA sequences and two kinds of aptamers against platelet-derived growth factor (PDGF-BB) and thrombin, respectively. After the capture of the target and the sandwich-type reaction, DNA-labeled AgNPs were captured to form the AgNPs aggregates on the electrode surface through the in situ hybridization of DNA. A remarkably amplified electrochemical signal was observed by differential pulse stripping voltammetry (DPSV). The sensitivity of this in situ hybridization-induced formation of AgNPs aggregates was calculated to be 10 orders of magnitude higher than that of the single AgNP nanolabel. Moreover, the DNA-induced assembly of AuNPs was employed for protein kinase activity analysis (Figure 1B) [38]. In this paper, the Zr^4+^-labeled phosphorylated peptide could capture DNA-modified AuNPs (DNA-AuNPs) via the coordination interaction between the phosphate groups in DNA and Zr^4+^ ions Then, DNA-AuNPs polymeric networks were formed in situ by DNA hybridization on the electrode surface. The conductive and negative charged networks could accommodate a large amount of [Ru(NH_3_)_6_]^3+^ ions by the electrostatic interactions. The current intensity was dramatically enhanced, and a low detection limit and a wide linear range were achieved.

At the same time, other DNA-based assembly nanotechnologies including enzyme-aided and enzyme-free methods have also gained attractive attention. For example, Yu et al. developed a cascade signal amplification platform through integrating duplex-specific nuclease (DSN)-assisted target recycling with CHA reaction for the detection of microRNA-141 (miR-141) (Figure 2) [39]. During the DSN-assisted target recycling amplification, one miR-141 extracted from human breast cancer cells could induce the production of massive DNA connectors, which would trigger the next CHA reaction. Then, the AuNP hot spots were self-assembled into networks on the H_2_-immobilized electrode surface. Numerous positively charged RuHex ions were captured by the anionic phosphate backbone of DNA duplex, finally resulting in a significant amplification in the electrochemical signal. Moreover, two-input AND and INHIBIT (INH) molecular logic gates were fabricated to analyze miRNAs. As one enzyme-free isothermal alternative, HCR avoids the restriction of precise dependence of pH and temperature in enzyme-mediated methods and has been used to detect miRNA and others [40,41]. For example, Yuan et al. developed an electrochemical biosensor for the simultaneous detection of multiple miRNAs, in which the DNA-modified magnetic nanoprobes loaded with two different electroactive molecules were bound with the products of HCR [42]. However, in this method, one copy of miRNA only triggered one copy of polymeric HCR product. To further improve the sensitivity of the HCR-based method, Miao et al. reported an electrochemical method for miRNA detection based on the analyte-triggered nanoparticle localization on the electrode in combination with HCR amplification [43]. In this work, miRNA induced the opening of a hairpin on the electrode-immobilizing tetrahedral DNA for the capture of HCR-H0-modified AuNPs. Free HCR-H0 strands on the surface of AuNPs could further induce in situ hairpin polymerization. Subsequently, numerous AgNPs were assembled on the electrode, generating a sharp stripping current peak during the solid-state Ag/AgCl reaction. Although DNA-modified NPs as the sensing units have been widely used in electrochemical assays, extensive and complicated conjugation steps increase the complexity and cost of the assays.

To overcome the shortcoming of DNA label-based experiments, label-free and nanomaterials-based signal amplification strategies have attracted more interest. As is known to us, the polymeric products generated by DNA self-assembly can be used as the templates for the assembly of nanomaterials through the electrostatic interaction or in situ metallization [44,45]. For this view, Li et al. demonstrated that the positively charged AuNPs could electrostatically assemble onto the double-helix of HCR products to amplify the electrochemical signal [46]. It was reported that a cytosine (C)-rich DNA sequence can be used as the template to prepare silver nanoclusters (AgNCs) that showed excellent electrocatalytic ability and redox property [47,48]. Yang et al. developed a label-free electrochemical method for the detection of miRNA (miRNA-199a) based on the in situ DNA-templated synthesis of AgNCs [49]. As displayed in Figure 3, when miRNA-199a was hybridized with the template probe, the target-assisted polymerization nicking reaction (TAPNR) amplification was initiated, and massive intermediate sequences were generated to bind with the secondary DNA probes on the electrode. Then, the HCR amplification was triggered by the surface-tethered intermediate sequences. In this process, numerous C-rich loop DNAs were formed in the dsDNA polymers. In the presence of AgNO_3_ and NaBH_4_, a large amount of AgNCs were produced by using C-rich loop DNAs as the templates. Similar to AgNPs, the AgNCs could generate a dramatically enhanced current response. Up to now, integrating HCR with the in situ formation of AgNCs has also been used to develop electrochemical biosensors for the detection of methyltransferase activity, Pb^2+^ and Type b3a2 [50,51,52]. In addition, AgNCs generated in the polymeric HCR products exhibit excellent ECL property and have been utilized to detect HATs activities [53]. In addition, double-stranded DNA (dsDNA) can be employed as the template for the preparation of copper nanoclusters (CuNCs). Zhao et al. reported an electrochemical method for protein detection based on the HCR-assisted formation of CuNCs [54]. The formed CuNCs could release numerous Cu^2+^ ions by acid dissolution, thus catalyzing the oxidation of o-phenylenediamine by O_2_ and leading to the strong electrochemical signal. With exonuclease T7 triggered targets recycling and HCR amplification, Wang et al. prepared tree-like overlapping and branching Y-shaped dsDNA for the precise in situ growth of CuNCs [55].

### 2.2. Metal Ion–Ligand Coordination

Organic ligands modified on the surface of NPs can bind metal ions with different stability constants. The formation of the ligand–metal–ligand complex can induce the aggregation of NPs. In 2015, Wei et al. first proposed the concept of converting liquid-phase colorimetric assay into enhanced surface-tethered electrochemical analysis [28]. Based on the strategy, Hg^2+^ was sensitively detected as the model analyte. As shown in Figure 4A, cysteamine-capped AgNPs were prepared and modified with thymine-1-acetic acid, in which thymine (T) could be specifically coordinated with Hg^2+^ by the formation of a T-Hg^2+^-T bond. Meanwhile, the gold electrode was modified with thymine-1-acetic acid. Hg^2+^ was captured by thymine modified on the electrode surface, which allowed for the attachment of thymine-functionalized AgNPs (Ag-T) nanoprobes. Subsequently, surface-bound Ag-T nanoprobes could induce more Hg^2+^ and nanoprobes to be immobilized on the electrode, thus leading to the formation of Ag-T-based nanostructures. Finally, a strong and well-defined electrochemical signal was attained. The detection limit of the proposed electrochemical sensor was approximately two orders of magnitude lower than that of the AgNPs-based colorimetric assay of Hg^2+^. Similarly, Cu^2+^ and Cr^3+^ were selectively detected through their coordination with 4-mercaptobenzoic acid and 3-mercaptopropanoic acid, respectively [56,57]. Recently, Gu et al. employed T-functionalized upconversion nanoparticles (UCNP) as sensing units (T-UCNP) to develop an ECL sensor for the analysis of Hg^2+^ [58]. After the target-induced aggregation, multiple UCNPs were assembled on the electrode surface, and an amplified ECL signal was generated.

Biomolecules that can specifically interact with metal ions can also been detected by this method. For example, Zhao et al. reported an electrochemical method for the chiral recognition of D-/L-tryptophan (Trp) based on the Cu^2+^-assisted NPs aggregation [59]. As shown in Figure 4B, D-Trp functionalized AuNPs (D-Trp-AuNPs) were used to modify the glass carbon electrode. As a result of the high binding constant with NPs and D-Trp, Cu^2+^ could induce more electroactive Au@Ag NP to assemble on the D-Trp-AuNPs-modified electrode. In the presence of D-Trp, more Au@Ag NPs networks were formed on the electrode surface, and a strong DPV was observed. In addition, Wang et al. reported the detection of lipopolysaccharide based on the Cu^2+^-induced formation of AuNPs aggregates as the signal labels on the electrode surface, in which anionic groups were coordinated with Cu^2+^ ions to induce the nanoparticle aggregation [60].

### 2.3. Metal–Thiol and Boronate Ester Interactions

Phenylboronic acid (PBA) and its derivates can bind with α-hydroxycarboxylate acids (such as citrate and tartrate) and o-diphenol/diol-containing species (such as catechol derivatives, nucleosides, and glycoproteins) via the formation of a covalent boronate ester bond [61]. Such interactions have been introduced into the design of various biosensors [62,63,64]. Capping reagents play an important role in the enhancement of the stability and solubility of nanomaterials [65,66]. Among them, trisodium citrate is the most frequently used reagent for the synthesis of negatively charged AgNPs and AuNPs. Unlike DNA, the ribose group in RNA contains an intact cis-diol structure in the pentose ring at the 3′-terminal, which can react with PBA to form a boronate ester covalent bond. Our group reported a label-free electrochemical method for the detection of miRNAs based on the in situ aggregation of AgNPs [67]. As shown in Figure 5, after the hybridization, the exposed cis-diol moiety in the ribose of the captured miRNA was derivatized by 4-mercaptophenylboronic acid (MPBA). Next, the thiol group of MPBA could grasp citrate-capped AgNPs via the formation of a Ag-S bond. Then, the surface-bound AgNPs could recruit more MPBA and AgNPs from solution through the formation of boronate ester and Ag-S bonds, finally resulting in the in situ generation of MPBA-AgNPs-based networks on the electrode surface. Based on the amplified electrochemical signal, miRNA-21 was sensitively determined with a detection limit down to 20 aM.

Tyrosinase can catalyze the oxidation of tyrosine residue in the substrate peptide and transform monophenol into o-diphenol, which could be recognized by MPBA. Based on the MPBA-AgNPs-based networks, our group developed two sensitive electrochemical strategies for the determination of protein kinase activity [68]. As shown in Figure 6, after the hydroxyl of tyrosine residue was phosphorylated in the presence of tyrosine kinase (Src) and 5-[-thio] triphosphate (ATP-S), the thiophosphate peptide could bind to AgNPs through the formation of a Ag-S bond. Eventually, under the MPBA-assisted in situ assembly of AgNPs, the nanoarchinectures were formed on the electrode. In another strategy, after the oxidation of tyrosine residue, MPBA reacted with the o-diphenol moiety, and then, AgNP was captured by MPBA through the Ag-S interaction, leading to the generation of AgNPs-based networks. However, once the tyrosine residue was phosphorylated, the oxidation and assembly process would be blocked, resulting in the decrease in the current intensity. Moreover, glycan on the surface of glycoprotein can also react with MPBA to induce the formation of MPBA-AgNPs-based networks on the electrode surface, thus allowing for the development of electrochemical glycoprotein aptasensors [69].

Benzene-1,4-dithiol (BDT) with two thiol groups can be used as the connector for the assembly of AgNPs. Based on the BDT-induced in situ formation of AgNPs networks, Hou et al. constructed a modification-free amperometric biosensor for the detection of wild-type p53 protein [70]. As displayed in Figure 7, a dsDNA probe containing two consensus sites was employed to modify the gold electrode for the capture of wild-type p53 protein. After the binding between the probe and protein, the thiol and amine groups on the surface of p53 protein bind to AgNPs via the formation of Ag-S and Ag-N bonds. In the presence of BDT, more AgNPs were in situ assembled on the electrode surface to form the networks for signal amplification. This method has been successfully used to detect wild-type p53 protein in cell lysates with satisfactory results.

### 2.4. Peptide-Induced Assembly of Nanoparticles

As a result of the diversity of structural units (amino acids), peptides can be synthesized with a specific sequence and used as the aptamer for the capture of the target. Moreover, a peptide with positive charges can induce the aggregation of negatively charged citrate-capped AuNPs and AgNPs via the electrostatic interactions. In this process, the peptide probe plays two roles (the target binder and the NPs aggregation inducer). Once the peptide was bound to the target, it would lose the ability to trigger the aggregation of NPs. Based on this fact, AuNPs and AgNPs-based liquid-phase colorimetric assays have been converted into surface-tethered electrochemical electrochemical assays [71]. For example, the tripeptide (Arg-Pro-Arg) with two positively charged arginine residues could lead to the AuNPs aggregation. Dipeptidyl peptidase-IV (DPP-IV) can induce the hydrolyzation of the peptide, thus preventing the aggregation of AuNPs. DPP-IV activity was determined by the colorimetric and electrochemical methods based on the peptide-induced AuNPs aggregation [72].

PrP(95−110) with an amino acid sequence of THSQWNKPSKPKTNMK was identified as the receptor of small amyloid-β (Aβ) oligomer (AβO) with high specificity and affinity [73,74,75,76]. Our group found that PrP(95−110) could induce the AuNPs or AgNPs aggregation with a color change [77,78]. For this view, we further developed an electrochemical platform for the detection of AβO based on the in situ formation of AgNPs networks for signal amplification [78]. As shown in Figure 8A, the adamantine (Ad)-modified PrP(95−110) could be attached on the surface of AgNPs to result in their aggregation. Based on the host–guest interaction, the formed Ad-PrP(95−110)-AgNPs networks were tethered on the β-CD (β-cyclodextrin)-modified electrode surface, thus producing an amplified electrochemical signal. However, in the presence of AβO, the binding of AβO to the peptide hindered the interaction of the peptide and AgNPs, thus leading to the reduced magnitude of aggregation on the electrode and decreasing the electrochemical signal from the oxidation of AgNPs. 

As a hormone produced by placenta, human chorionic gonadotropin (hCG) is recognized as an important indicator for pregnancy and several cancers. Our group designed an electrochemical biosensor for hCG detection with a dual-functional peptide probe (PPLRINRHILTR) [79]. As shown in Figure 8B, the positively charged hCG-binding peptide used as the sensing unit can induce the aggregation of AuNPs via the electrostatic interactions and facilitate the in situ formation of AuNPs, which assemble on the electrode surface. The formed networks could significantly reduce the charge transfer resistance. However, in the presence of hCG, the stable complex of the peptide probe and hCG lost the coagulating ability toward AuNPs. The amount of AuNPs assemblies on the electrode was reduced, and the charge transfer resistance was intensified. This method based on the electrochemical impedance technique achieved the determination of hCG with a detection limit of 0.6 mIU/mL. By using AgNPs as the redox probes for a well-defined and amplified electrochemical signal, hCG could be sensitively measured by linear-sweep voltammetry (LSV) [80].

As one type of essential structural molecules, peptides with excellent self-assembly and molecular recognition ability can be self-assembled into various nanostructures. Moreover, through the careful encoding of peptide with a binding or reactive site, peptides can co-assemble with additional nanomaterials with unique optical and chemical properties into functional hybrid superstructures. Recently, Han et al. reported the co-assembly of peptides and carbon nanodots (CNDs) (Pep/CND) based on the π−π stacking between tyrosine residues and CNDs (Figure 9A) [81]. The peptide and CNDs endowed Pep/CND co-assembly with the recognition capability and the catalytic activity, respectively. They further applied the Pep/CND co-assembly to construct an electrochemical method for the detection of transglutaminase 2 (TG2). As shown in Figure 9B, TG2 catalyzed the ligation of peptide P2 and peptide P1 on the electrode surface. Next, CNDs was bound to P2, subsequently triggering the co-assembly of a plenty of P3 and CNDs because the tyrosine was located at one terminal. The large amount of CNDs with catalytic ability could catalyze the redox reaction between H_2_O_2_ and 3,3′,5,5′-tetramethylbenzidine (TMB), resulting in the enhancement of electrochemical signal for the sensitive detection of TG2.

### 2.5. SA–Biotin Interaction

SA is a tetrameric protein that can bind to four biotin molecules with high binding affinity in a wide pH range (K_d_ = 10^−15^ M) [82,83]. The specific and strong interaction is always utilized for the conjugation of antibodies or nucleic acids with enzymes or nanomaterials for signal output. It has been reported that aromatic phenylalanine (Phe) and its derivates can self-assemble into various nanostructures through the modulation of different parameters. Our group found that the biotinylated Phe (biotin-Phe) monomers can self-assemble into monodispersed biotin–Phe nanoparticle (biotin–FNP) by controlling the pH value [84]. Then, an impedimetric biosensor for the determination of caspase-3 activity and evaluation of cell apoptosis was based on the in situ assembly of biotin–FNP in the presence of SA [85]. As shown in Figure 10, tetrameric SA protein can be captured by the biotinylated DVED-containing peptide. Then, free biotin-binding sites of SA on the electrode surface allowed for the anchor of biotin–FNP and SA, finally leading to the formation of SA–biotin–FNP networks on the electrode surface. The direct electron transfer between [Fe(CN)_6_]^3−/4−^ and the electrode was seriously hindered. In the presence of caspase-3, the cleavage of peptide prevented the binding of SA and the follow-up formation of biotin–FNP networks on the electrode surface. The electron transfer resistance (R_et_) was inversely proportional to the concentration and activity of caspase-3. The method was further employed to develop aptasensors for impedimetric detection of miRNAs and small molecules by competitive reactions [84,86].

## 3. In Situ Assembly of Small Molecules and Biomolecules

In DNA assembly-based electrochemical methods, the DNA assembly unit is usually conjugated with an electroactive molecule for signal output. Actually, the amount of electroactive molecule in the assembly product is low, and the insulation property of DNA and its assembly may decrease the detection performances. Recently, small molecules and biomacromolecules are proposed as the assembly units to form nanostructures on the electrode surface for improving the detection sensitivity. For example, inspired by the polymerization reaction, Hu et al. has reported the application of surface-initiated electrochemically mediated atom-transfer radical polymerization (SI-eATRP) as an amplification strategy for the electrochemical biosensing of different targets, including double-stranded DNA (dsDNA) and protein kinase activity [87,88]. Typically, they demonstrated the de novo growth of a polymers (dnGOPs)-based electrochemical biosensor for the detection of target DNA (tDNA) through SI-eATRP [89]. The principle of this method was illustrated in Figure 11A. Peptide nucleic acid (PNA) probes with a neutrally charged N-(2-aminoethyl)glycine units-composed backbone were immobilized on the electrode to specifically capture tDNA. After hybridization, phosphate groups with high density could bind to Zr^4+^ and α-bromophenylacetic acid (BPAA) through the phosphate-Zr^4+^-carboxylate coordination chemistry. With the aid of a constant potential, the SI-eATRP was triggered, and numerous electroactive ferrocenylmethyl methacrylate (FMMA) monomers were polymerized into long polymeric chains on the electrode surface. Since the electrochemical response was greatly improved, tDNA was sensitively and selectively detected with a detection limit of 0.072 fM. However, the utilization of Cu^2+^ ions as catalysts may interfere with the next electrochemical experiments because of the non-specific interaction with nucleic acids and the electrochemical deposition of metal on the electrode. For this consideration, they further explored novel surface-initiated electrochemically controlled reversible-addition-fragmentation-chain-transfer (SI-eRAFT) polymerization without the use of transition metal ions for assays [90,91,92]. For instance, they reported a signal-amplified electrochemical sensing of thrombin activity by SI-eRAFT polymerization [93]. As shown in Figure 11B, after the thrombin-specific substrate peptide (Tb peptide) was cleaved, 4-cyano-4-(phenylcarbonothioylthio)pentanoic acid (CPAD) was bound to the peptide by the formation of the carboxylate–zirconium–carboxylate complexes. The initiation of SI-eRAFT resulted in the polymerization of a large number of FMMA on the electrode surface, leading to the significant increase in the current intensity. The results showed that the SI-eRAFT-based amplification strategy held great promise in the sensitive analysis of biomolecules. Recently, an in situ initiated ring-opening polymerization signal amplification strategy was also integrated with an electrochemical biosensor for the detection of CYFRA 21-1, which is a specific biomarker for non-small cell lung cancer [94]. In addition to the covalent bond for the polymerization of signal molecules, a peptide can self-assemble into various stable nanostructures via non-covalent interaction including hydrogen bonding, hydrophobic interaction, electrostatic interaction, non-specific Van der Waals and π–π stacking. Inspired by DNA assembly techniques, Huang et al. reported a signal amplification strategy based on the in situ self-assembly of peptides for the determination of AβO [95]. After being captured by the peptide CP4-PrP(95–110), the captured AβO could trigger the in situ self-assembly of the amphiphilic C16-GGG-PrP(95–110)-Fc peptide on the surface of the electrode under mild conditions. The accumulation of numerous Fc molecules generated a significantly amplified signal.

Generally, the efficiency of enzyme-catalyzed hydrolysis or modification of substrate peptide or nucleic acid may be hampered due to the steric hindrance and low freedom of the substrate. Thus, it is useful to integrate the homogeneous assay with the surface-tethered electrochemical analysis, which can retain the high efficiency of the enzymatic reaction and the high selectivity of the interfacial analytical method. Our group reported an electrochemical caspase-3 biosensor by conversing a homogeneous assay into a surface-tethered electrochemical analysis based on the SA–biotin interaction [96]. As shown in Figure 12, SA molecules could co-assemble with the peptide substrates (biotin–GDEVDGK–biotin) to form (SA–biotin–GDEVDGK–biotin-SA)_n_ aggregates on the SA-modified electrode surface. The in situ performed aggregates significantly blocked the electron transfer of [Fe(CN)_6_]^3−/4−^ and increased the R_et_. However, after the homogeneous cleavage of the peptide by caspase-3, the amount of intact biotin–GDEVDGK–biotin decreased, and the product of biotin-labeled pieces further competed with the peptide substrate to bind SA, leading to the suppression of the in situ assembly of biotin–GDEVDGK–biotin and SA. DNA probes can be employed as the assembly units for the detection of miRNAs and enzyme activities. Our group also developed a DSN-based electrochemical biosensor for the analysis of miRNAs by integrating homogeneous enzymatic reaction with surface-tethered electrochemical analysis [97]. In the work, the biotinylated DNA (biotin–DNA–biotin) can trigger the in situ co-assembly of DNA and SA on the electrode. The insulating (SA–biotin–DNA–biotin)_n_ assemblies could hamper the electron communication between [Fe(CN)_6_]^3−/4−^ and the electrode. However, when the biotinylated DNA was hybridized with the target miRNA, the biotin–DNA–biotin in the dsDNA would be hydrolyzed by DSN, resulting in the hybridization–enzymolysis cycle and the generation of abundant biotin–DNA fragments. The released fragments could further compete with biotin–DNA–biotin to bind SA and thus reduced the amount of (SA–biotin–DNA–biotin)_n_ assemblies. In addition, our group also reported a (SA–biotin–DNA–biotin)_n_ networks-based electrochemical biosensor for the detection of telomerase activity in cancer cells [98]. The primer was extended by telomerase to generate many (TTAGGG)n repeats on the electrode surface. The elongated primer could hybridize with its complementary sequence biotin–DNA–biotin, subsequently triggering the in situ co-assembly of biotin–DN–biotin and SA into (SA–biotin–DNA–biotin)_n_ networks. Numerous phosphate groups in networks blocked the access of negatively charged [Fe(CN)_6_]^3−/4−^ to the electrode surface, resulting in the increase in R_et_. Based on the signal-amplified strategy, telomerase extracted from two HeLa cells could be readily determined.

The applications and analytical performances of different assembly strategies are summarized in Table 1.

## 4. Conclusions

Liquid-phase aggregation-based assays, such as colorimetric and fluorescence assays, are simple and convenient. In contrast to the homogeneous analysis, heterogeneous interface assays are considered as more sensitive and accurate tools for biochemical detection. Nanomaterials in aggregation state may exhibit different optical and chemical properties compared to the monodispersed state. In this review, we summarize the advancements in the in situ assembly of nanomaterials for the signal output and amplification of electrochemical biosensors. The proposed strategies by converting liquid-phase aggregation-based assay into sensitive surface-tethered electrochemical analysis have exhibited more advantageous performances and have shown promising applications. It is expected that new assembly methods and units would be more effective and abundant to achieve highly sensitive and specific detection.

## Figures and Tables

**Figure 1 nanomaterials-11-03307-f001:**
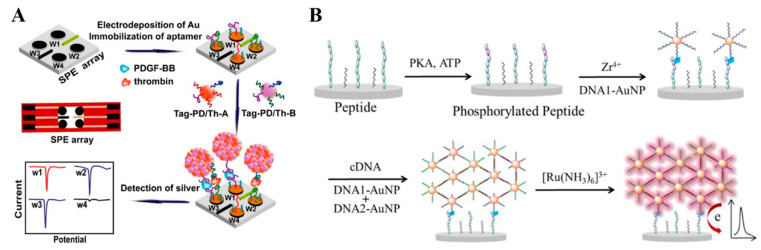
(**A**) Schematic representation of the principle of the dnGOPs-based electrochemical detection of DNA [37]. Copyright 2014 American Chemical Society. (**B**) Schematic representation of DNA-AuNPs assembled polymeric network amplified electrochemical biosensor for kinase activity detection [38]. Copyright 2014 American Chemical Society.

**Figure 2 nanomaterials-11-03307-f002:**
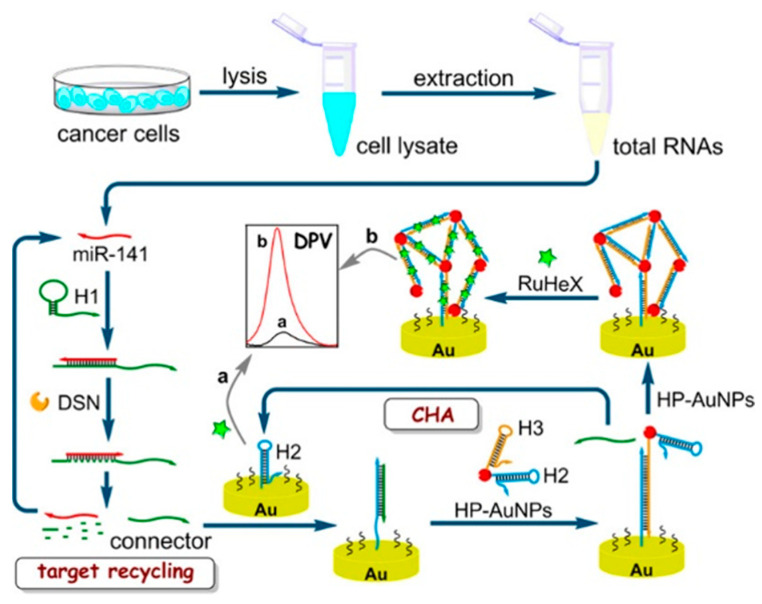
Schematic representation of cascade amplification of DSN-assisted target recycling and CHA reaction and in situ self-assembly of AuNP networks on electrodes for label-free electrochemical detection of miR-141 in signal-on mode [39]. Copyright 2018 American Chemical Society.

**Figure 3 nanomaterials-11-03307-f003:**
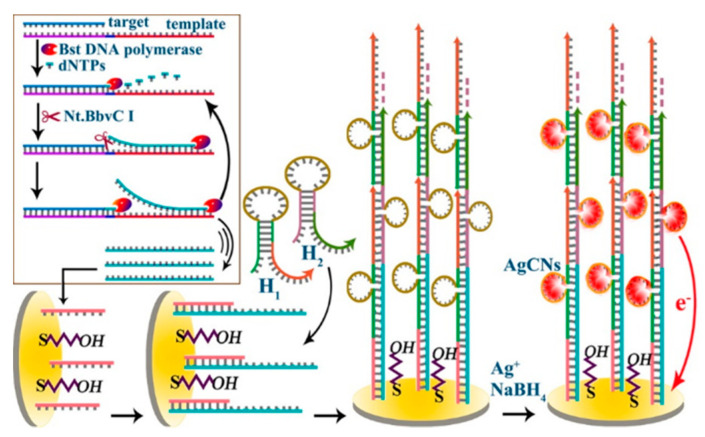
Schematic representation of ultrasensitive and label-free electrochemical detection of miRNA-199a based on in situ generated AgNCs by coupling TAPNR with HCR amplifications [49]. Copyright 2015 American Chemical Society.

**Figure 4 nanomaterials-11-03307-f004:**
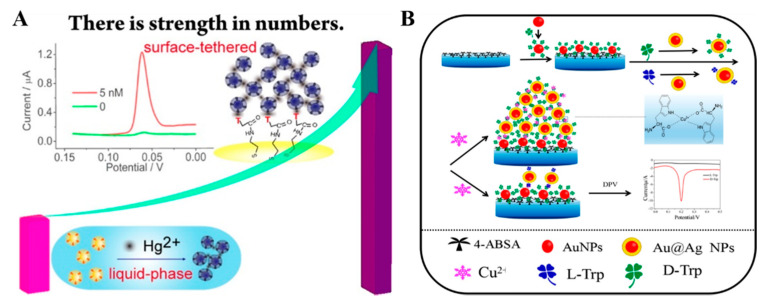
(**A**) Schematic representation of the electrochemical Hg^2+^ sensor based on Ag-T nanoprobes [28]. Copyright 2015 American Chemical Society. (**B**) Schematic representation of electroactive Au@Ag NP assembly driven signal amplification for ultrasensitive chiral discrimination of D-/L-Trp [59]. Copyright 2019 American Chemical Society.

**Figure 5 nanomaterials-11-03307-f005:**
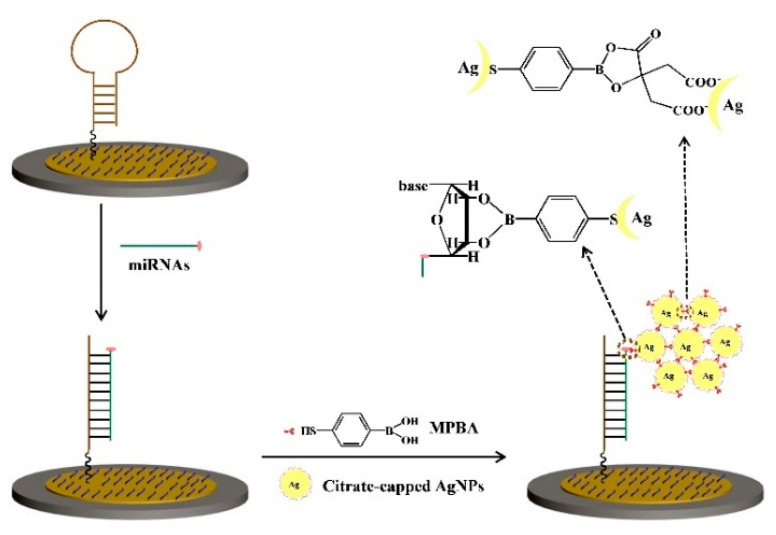
Schematic representation of the proposed electrochemical strategy for miRNAs detection based on MPBA-induced in situ formation of AgNPs aggregates as labels [67]. Copyright 2017 Elsevier B.V.

**Figure 6 nanomaterials-11-03307-f006:**
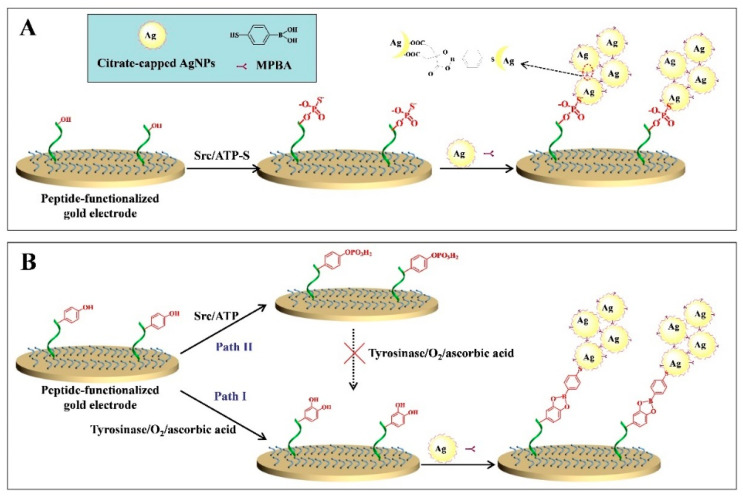
Schematic representation of the proposed electrochemical strategies for protein kinase detection based on the in situ formation of the AgNPs aggregates as labels. In the first design (**A**), ATP-S was used as the co-substrate. In the second design (**B**), ATP was used as the co-substrate, and tyrosinase was used to convert monophenol into o-diphenol [68]. Copyright 2017 Elsevier B.V.

**Figure 7 nanomaterials-11-03307-f007:**
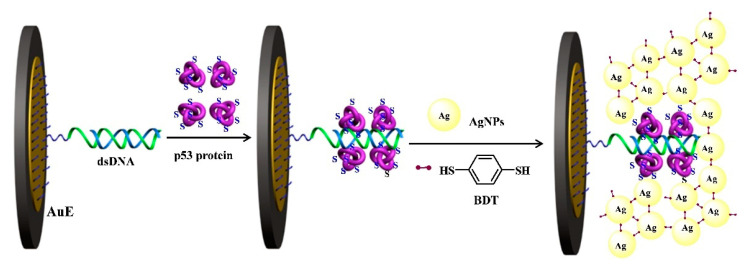
Schematic representation of the biosensor for the detection of wild-type p53 protein by the in situ formation of AgNPs networks for signal amplification [70]. Copyright 2020 Elsevier B.V.

**Figure 8 nanomaterials-11-03307-f008:**
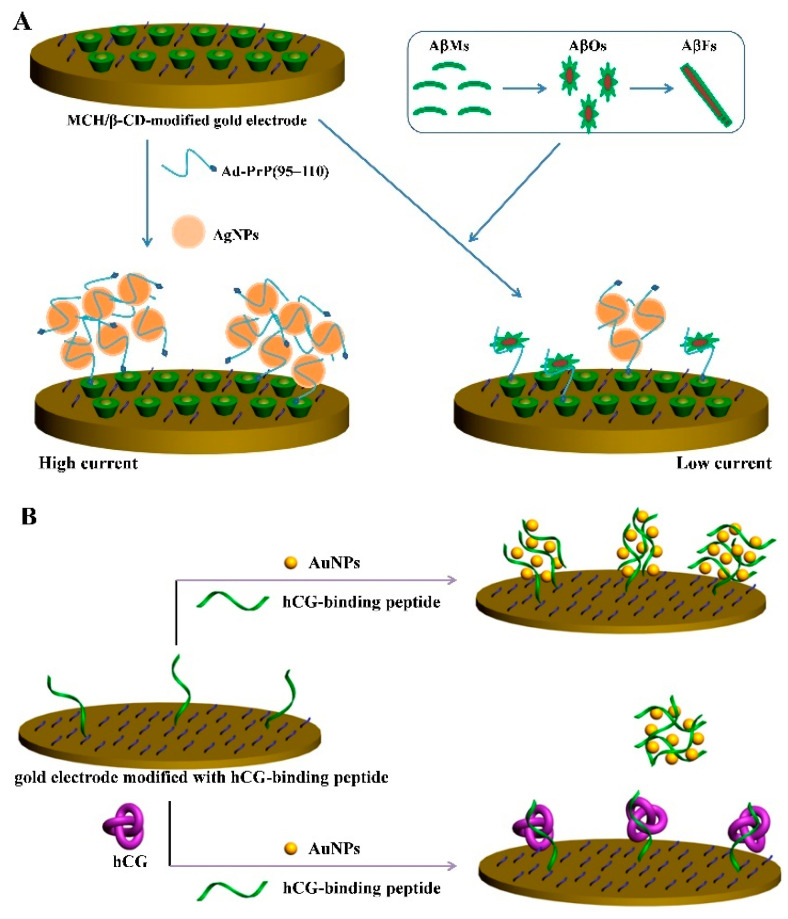
(**A**) Schematic representation of the electrochemical method for the selective detection of AβOs using AgNPs as the redox reporters and Ad-PrP(95−110) as the receptor [78]. Copyright 2016 American Chemical Society. (**B**) Schematic illustration of the electrochemical method for hCG detection using a peptide probe as the receptor of hCG and the inducer of AuNPs assembly [79]. Copyright 2017 Elsevier B.V.

**Figure 9 nanomaterials-11-03307-f009:**
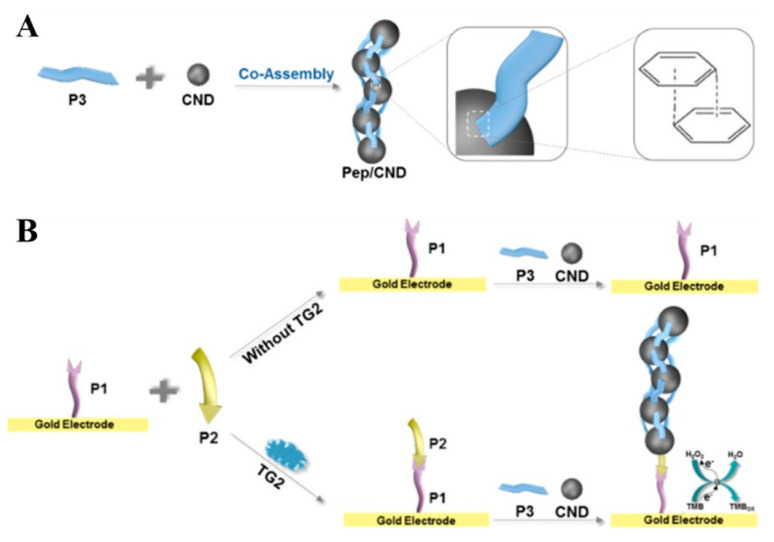
Schematic representation of (**A**) the co-assembly of P3 and CNDs and (**B**) the principle of the analysis of TG2 [81]. Copyright 2021 American Chemical Society.

**Figure 10 nanomaterials-11-03307-f010:**
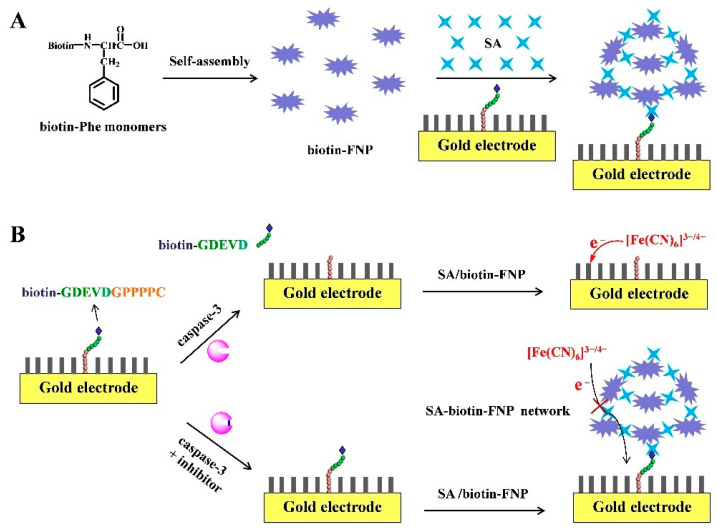
(**A**) Self-assembly of biotin–Phe monomers into biotin–FNP and the in situ formation of SA–biotin–FNP networks on the electrode surface. (**B**) Schematic representation of the biosensor for assay of caspase-3 activity via the signal amplification by SA–biotin–FNP networks [85]. Copyright 2020 Elsevier B.V.

**Figure 11 nanomaterials-11-03307-f011:**
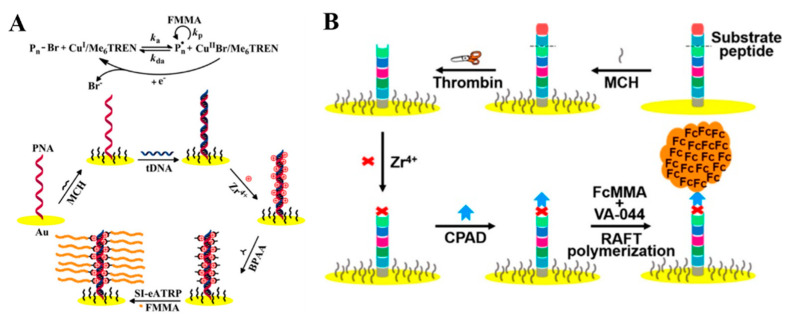
(**A**) Schematic representation of the principle of the dnGOPs-based electrochemical detection of DNA [89]. Copyright 2017 American Chemical Society. (**B**) Schematic representation of “signal-on” electrochemical biosensing of thrombin activity [93]. Copyright 2020 American Chemical Society.

**Figure 12 nanomaterials-11-03307-f012:**
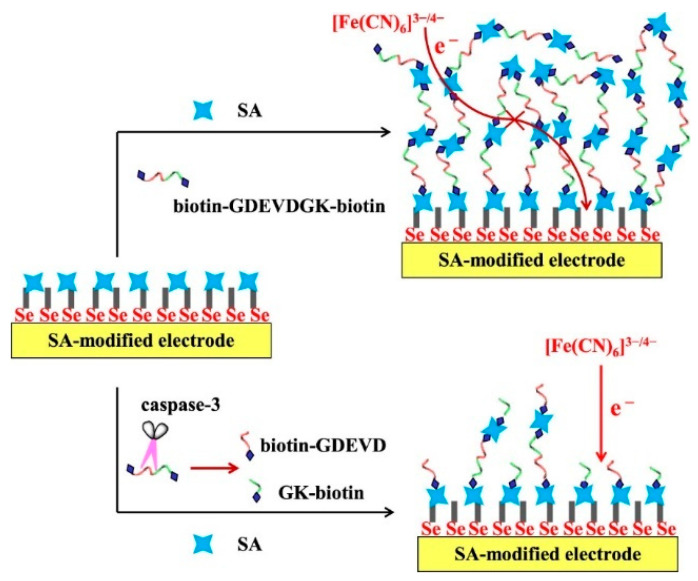
Schematic representation of the surface-tethered electrochemical analysis for caspase-3 detection [96]. Copyright 2021 American Chemical Society.

**Table 1 nanomaterials-11-03307-t001:** Analytical performances of different electrochemical biosensors based on the in situ assembly of nanomaterials and molecules for the signal enhancement.

Interaction	Nanomaterials	Target	Detection Range	Detection Limit	Ref.
DNA hybridization	DNA-AuNPs	lysozyme	1 pM–1 nM	0.32 pM	[36]
	DNA-AgNPs	PDGF and thrombin	5 pg/mL–1000 ng/mL	1.6 pg/mL	[37]
	DNA-AuNPs	PKA	0.03–40 U/mL	0.03 U/mL	[38]
	DNA-AuNPs	miR-141	0.1 fM–10 nM	25.1 aM	[39]
	DNA-CeO_2_	VEGF	1 fg/mL–0.1 ng/mL	0.27 fg/mL	[41]
	Thi-modified DNA-Fe_3_O_4_ NPs and Fc-CHO-modified DNA-Fe_3_O_4_ NPs	miR-141 and miR-21	1 fM–1 nM	0.44 fM for miR-141 and 0.46 fM for miR-21	[42]
	DNA-AgNPs	miR-17-5p	100 aM–100 pM	2 aM	[43]
DNA-based electrostatic interaction	Ag@Au CSNPs	Hg^2+^	10 pM–2.5 nM	3.6 pM	[44]
	CTAB-capped AgNPs	PSA	0.1 pg/mL–75 ng/mL	0.033 pg/mL	[45]
	AuNPs	DNA	15 pM–1.0 nM	2.6 pM	[46]
DNA-based in situ metallization	AgNCs	DNA	0.2 fM–1 pM	0.16 fM	[48]
	AgNCs	miR-199a	1.0 fM–0.1 nM	0.64 fM	[49]
	AgNCs	methyltransferase	0.02–10 U/mL	0.0073 U/mL	[50]
	AgNPs	Pb^2+^	1 pM–100 nM	0.24 pM	[51]
	AgNPs	Type b3a2	10 fM–10 nM	0.56 fM	[52]
	AgNCs	HAT	0.5–100 nM	0.49 nM	[53]
	CuNPs	folate receptor	0.01–100 ng/mL	3 pg/mL	[54]
	CuNCs	miR-21	10 pM–0.1 fM	10 aM	[55]
Metal ion–ligand coordination	thymine-modified AgNPs	Hg^2+^	50 pM–50 nM	5 pM	[28]
	MBA-modified AgNPs	Cu^2+^	0.1–100 nM	0.08 nM	[56]
	MPA-modified AgNPs	Cr^3+^	200–5000 ppb	278 ppb	[57]
	Thymine-modified UCNPs	Hg^2+^	10 pM–100 nM	0.4 pM	[58]
	Au@Ag NPs	D-tryptophan	5 pM–1 nM	1.21 pM	[59]
	L-cysteine-modified AuNPs	lipopolysaccharide	1.0–10 pg/mL	0.033 pg/mL	[60]
Metal–thiol and boronate ester interactions	citrate-capped AgNPs	tyrosinase	0.001–0.5 mU/mL	0.1 mU/mL	[61]
	citrate-capped AgNPs	thrombin	0.025–5 ng/mL	0.02 ng/mL	[61]
	citrate-capped AgNPs	H_2_O_2_	1 nM–0.6 μM	Not reported	[62]
	citrate-capped AgNPs	miR-21	0.1–50 fM	20 aM	[67]
	citrate-capped AgNPs	tyrosine kinase	0.1–25 ng/mL	0.1 ng/mL	[68]
	citrate-capped AgNPs	PSA	0.5–200 pg/mL	0.2 pg/mL	[69]
	citrate-capped AgNPs	wild-type p53	0.1–100 pM	0.1 pM	[70]
Peptide-induced assembly	citrate-capped AuNPs	PKA	0.01–1 U/mL	20 mU/mL	[71]
	citrate-capped AuNPs	DPP-IV	0.001–0.5 mU/mL	0.55 µU/mL	[72]
	citrate-capped AgNPs	AβO	0.01–200 nM	6 pM	[73]
	citrate-capped AgNPs	AβO	20 pM–100 nM	8 pM	[78]
	citrate-capped AuNPs	hCG	0.001–0.2 IU/mL	0.6 mIU/mL	[79]
	citrate-capped AgNPs	hCG	0.001–0.2 IU/mL	0.4 mIU/mL	[80]
	Carbon nanodots	transglutaminase 2	1 pg/mL–50 ng/mL	0.25 pg/mL	[81]
SA–biotin interaction	biotin-FNPs	aflatoxin B_1_	0.05–3 pg/mL	Not reported	[84]
	biotin-FNPs	caspase-3	1–125 pg/mL	1 pg/mL	[85]
	biotin-FNPs	miR-21	0.1–250 fM	0.1 fM	[86]
In situ assembly of small molecules and biomolecules	Fc derivate	DNA	1.0 fM–1.0 nM	0.47 fM	[87]
	Fc derivate	PKA	0–140 mU/mL	1.63 mU/mL	[88]
	Fc derivate	DNA	0.1 fM–0.1 nM	0.072 fM	[89]
	Fc derivate	DNA	10 aM–10 pM	3.2 aM	[90]
	Fc derivate	PKA	0–140 mU/mL	1.02 mU/mL	[91]
	Fc derivate	trypsin	25–175 μU/mL	18.2 μU/mL	[92]
	Fc derivate	thrombin	10–250 μU/mL	2.7 μU/mL	[93]
	Fc derivate	CYFRA 21-1	1 pg/mL–1 μg/mL	9.08 fg/mL	[94]
	Fc-labeled peptide	AβO	0.005–5 μM	0.6 nM	[95]
	(SA–biotin–peptide–biotin)_n_	caspase-3	0–50 pg/mL	0.2 pg/mL	[96]
	(SA–biotin–DNA–biotin)_n_	miR-21	0.01–2.5 fM	10 aM	[97]
	(SA–biotin–DNA–biotin)_n_	telomerase	20–5000 cells/mL	20 cells/mL	[98]

**Abbreviation:** AuNPs, gold nanoparticles; AgNPs, silver nanoparticles; PDGF, platelet-derived growth factor; PKA, protein kinase A; miR, microRNA; VEGF, vascular endothelial growth factor; Thi, thionine; Fc-CHO, ferrocene carboxaldehyde; CSNPs, core–shell nanoparticles; CTAB, cetyltrimethylammonium bromide; PSA, prostate specific antigen; AgNCs, silver nanoclusters; CuNPs, copper nanoparticles; CuNCs, copper nanoclusters; HAT, histone acetyltransferase; MBA, 4-mercaptobenzoic acid; MPA, 3-mercaptopropanoic acid; UCNPs, upconversion nanoparticles; DPP-IV, dipeptidyl peptidase-IV; AβO, amyloid-β oligomer; hCG, human chorionic gonadotropin; biotin–FNPs, biotin–phenylalanine-assembled nanoparticles; SA, streptavidin; Fc, ferrocene; CYFRA 21-1, cytokeratin19 fragment.

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
