# Peer review of "In Situ Assembly of Nanomaterials and Molecules for the Signal Enhancement of Electrochemical Biosensors"

_nanomaterials, 2021, doi:10.3390/nano11123307_

Round 1

Reviewer 1 Report

Authors overview the use of nanomaterials and other metal complexes for interaction with DNA, achieving signal amplification for electrochemical sensing. The manuscript is concisely written, the only issue I have is not citing more international literature.

Author Response

Comment: Authors overview the use of nanomaterials and other metal complexes for interaction with DNA, achieving signal amplification for electrochemical sensing. The manuscript is concisely written, the only issue I have is not citing more international literature.

Response: We thank the reviewer for his/her positive comment. More international references have been cited in the revised manuscript.

Reviewer 2 Report

The report is well written and covered a broad range of topics in in-situ nanomaterial assembly. I do not have major concerns with this review but would like to advise the author to diversify the referenced articles. The author self-cited 23 times out of 92 references. Although the author has made a significant contribution to the topic in discussion, I feel a more diverse representation of work from research groups around the world is needed.

Author Response

Comment: The report is well written and covered a broad range of topics in in-situ nanomaterial assembly. I do not have major concerns with this review but would like to advise the author to diversify the referenced articles. The author self-cited 23 times out of 92 references. Although the author has made a significant contribution to the topic in discussion, I feel a more diverse representation of work from research groups around the world is needed.

Response: We thank the reviewer for his/her positive comment. More international references have been cited in the revised manuscript.

Reviewer 3 Report

The paper reports on in-situ assembly of nanomaterials for sensors signal enhancement. The manuscript is well organized and clearly written. In my opinion, it could be published after minor revisions. Since some examples of DNA based biosensors reported are not really electrochemical devices, I'd rather suggest to change "electrochemical sensors" in "biosensors" in the paper title. the caption of Fig 2 should be improved.Please, briefly describe the steps reported in the Figure also in the caption. The caption of Fig 5 should be corrected. There is not any A label in the figure. Please, delete it from the caption. The bibliography could be completed by citing 

M. Terracciano, I. Rea, L. De Stefano, I. Rendina, G. Oliviero, F. Nici, S. D'Errico, G. Piccialli, N. Borbone, “Synthesis of mixed-sequence oligonucleotides on mesoporous silicon: chemical strategies and material stability” Nanoscale Research Letters2014, 9:317

and 

M. Terracciano, L. De Stefano, N. Borbone, J. Politi, G. Oliviero, F. Nici, M. Casalino, G. Piccialli, P. Dardano, M. Varra and I. Rea, “Solid phase synthesis of thrombin binding aptamer on macroporous silica for label free optical quantification of thrombin”, RSC Adv., 2016, 6, 86762 DOI: 10.1039/C6RA18401D.

Author Response

Comment: The paper reports on in-situ assembly of nanomaterials for sensors signal enhancement. The manuscript is well organized and clearly written. In my opinion, it could be published after minor revisions. Since some examples of DNA based biosensors reported are not really electrochemical devices, I'd rather suggest to change "electrochemical sensors" in "biosensors" in the paper title. the caption of Fig 2 should be improved. Please, briefly describe the steps reported in the Figure also in the caption. The caption of Fig 5 should be corrected. There is not any A label in the figure. Please, delete it from the caption. The bibliography could be completed by citing: M. Terracciano, I. Rea, L. De Stefano, I. Rendina, G. Oliviero, F. Nici, S. D'Errico, G. Piccialli, N. Borbone, “Synthesis of mixed-sequence oligonucleotides on mesoporous silicon: chemical strategies and material stability” Nanoscale Research Letters2014, 9:317 and M. Terracciano, L. De Stefano, N. Borbone, J. Politi, G. Oliviero, F. Nici, M. Casalino, G. Piccialli, P. Dardano, M. Varra and I. Rea, “Solid phase synthesis of thrombin binding aptamer on macroporous silica for label free optical quantification of thrombin”, RSC Adv., 2016, 6, 86762 DOI: 10.1039/C6RA18401D.

Response: We thank the reviewer for his/her positive comment. This review mainly focuses on the in-situ assembly of nanomaterials for the signal enhancement of electrochemical biosensors. We have changed “electrochemical sensors” into “electrochemical biosensors” in the paper title. We also revised the figure captions and cited more international references.

Reviewer 4 Report

In this paper, the authors discuss the design and application of electrochemical sensors based on the in-situ assembly of nanomaterials on the electrode surface for signal readout and amplification. Overall, the manuscript is well written and organized. However, there are some issues that should be resolved. Thus, I suggest the acceptance of this manuscript after the major revision.

1) The introduction part needs to be more focused on the performance/need of nanomaterials for the electrochemical sensors.

2) The section 3 (In-situ assembly of small molecules and biomolecules) does not match with the topic in this review article (in-situ assembly of nanomaterials). The authors need to delete this part or add other nanomaterials-based signal enhancement.

3) The authors need to prepare the table that summarizes the electrochemical sensors depending on the types of nanomaterials or interactions. In addition, the target molecules should be included with the detection sensitivities and range in the table.

Author Response

Comment 1: In this paper, the authors discuss the design and application of electrochemical sensors based on the in-situ assembly of nanomaterials on the electrode surface for signal readout and amplification. Overall, the manuscript is well written and organized. However, there are some issues that should be resolved. Thus, I suggest the acceptance of this manuscript after the major revision.

Response: We thank the reviewer for his/her positive comment.

Comment 2: The introduction part needs to be more focused on the performance/need of nanomaterials for the electrochemical sensors.

Response: We have revised the introduction and added some sentences to discuss the performance/need of nanomaterials for the electrochemical biosensors.

Comment 3: The section 3 (In-situ assembly of small molecules and biomolecules) does not match with the topic in this review article (in-situ assembly of nanomaterials). The authors need to delete this part or add other nanomaterials-based signal enhancement.

Response: In this review, we highlight the significant progress in the in-situ assembly of nanomaterials as the nanolabels for enhancing the performances of electrochemical biosensors. We also expand the range of the assembly units from nanomaterials to small organic molecules and biomolecules since some small organic molecules can be in-situ assembled into nanostructure networks as the signal labels and biomacromolecules with nanoscale sizes can also be used as the self-assembling building blocks to form diverse nanostructures for the development of electrochemical impedance biosensors.

Comment 4: The authors need to prepare the table that summarizes the electrochemical sensors depending on the types of nanomaterials or interactions. In addition, the target molecules should be included with the detection sensitivities and range in the table.

Response: The table (Table 1) has been added to summarize the performances of different electrochemical biosensors based on the in-situ assembly of nanomaterials for the signal enhancement.

Round 2

Reviewer 4 Report

The authors addressed well all the issues raised by the reviewers and I am satisfied with the modifications. So, I suggest the acceptance of this manuscript after the minor revision.

1) Regarding the response to comment 3, the title should be changed because authors intended to expand the range of the assembly units from nanomaterials to small organic molecules and biomolecules.

Author Response

Response: We thank the reviewer for his/her comments again. We have changed the title into "In-situ assembly of nanomaterials and molecules for the signal enhancement of electrochemical biosensors ".